# Effect of Surfactant Polyvinyl Pyrrolidone on the Properties of Microporous Carbon Nanospheres Reinforced Magnesium Matrix Composites

**DOI:** 10.3390/nano10112281

**Published:** 2020-11-17

**Authors:** Lin Jin, Yong-Zhen Yang, Jian-Feng Fan, Bing-She Xu

**Affiliations:** 1College of Mechanical and Vehicle Engineering, Taiyuan University of Technology, Taiyuan 030024, China; jinlin@tyut.edu.cn; 2Key Laboratory of Interface Science and Engineering in Advanced Materials (Ministry of Education), Taiyuan University of Technology, Taiyuan 030024, China; fanjianfeng@tyut.edu.cn

**Keywords:** magnesium matrix composites, microporous carbon nanospheres, polyvinyl pyrrolidone, interfacial bonds, mechanical properties

## Abstract

Microporous carbon nanospheres (PCNS)-reinforced magnesium (Mg) composites were prepared using polyvinyl pyrrolidone (PVP) as surfactant and PCNS as reinforcement. The influence of PVP treatment and the effectiveness of PCNS on the mechanical properties of Mg-based composites were investigated. The results show that the PCNS can enhance the properties of the Mg matrix. Moreover, the PVP can effectively improve the dispersion of PCNS in the Mg matrix but had a negative influence on the tensile properties of composites. The MgO films with high tensile strength were produced between matrix and reinforcement after removing PVP, which effectively promotes the interface compatibility and improves the properties of the composite. The tensile yield strength and specific strength of PCNS-reinforced Mg matrix composite exhibited 177 MPa and 102.4 × 103 N∙m/kg, respectively, which were 77% and 78% higher than those of the Mg matrix.

## 1. Introduction

As one of the lightest structural metals, magnesium (Mg) has many advantages of high specific strength, good damping, machinability and recyclability. Nanocarbon materials have been added into Mg matrix composite as reinforcements, which not only can maintain the outstanding properties of the Mg matrix but also overcome the intrinsic demerits of the Mg [1,2,3]. Microporous carbon nanospheres (PCNS), which are zero-dimensional structure and isotropy carbon nanomaterials, have a huge prospects for application in batteries, biomedicine, composite materials, energy storage and many other fields because of their good characteristics, such as low-density, good acid–base stability, low thermal expansion rate, nontoxic and potential in energy dissipation [4,5,6]. Meanwhile, PCNS is also proposed as an ideal reinforcement material for metal matrix composites. However, PCNS have poor dispersibility and compatibility in metal matrixes because of their high surface energy and specific surface area, limiting the proportion of nanocarbon reinforcement added in the Mg matrix directly and then impeding the improvement of composite properties.

In many preparation processes of metal composites, surface functionalization [7] could be used to functionalize the reinforcing phase, which can improve the dispersibility of the reinforcing phase and the compatibility of two phases effectively. At present, researchers have tried to use surfactants, such as cetyl trimethyl ammonium bromide (CTAB) [8], polyvinyl alcohol (PVA) [9], sodium dodecyl benzene sulfonate (SDBS) [10] and polyvinyl pyrrolidone (PVP) [11] to modify nanocarbon reinforcement. After this, the reinforcing phase is mixed with the Mg matrix and obtains the composites with homogeneously dispersed nanocarbon in the metal matrix. Finally, the properties of composites are improved. However, in fact, there are few surfactants to be used in the active Mg matrix. The selection of surfactant and formulation of the process should be based on the premise of avoiding Mg oxidation. Research has found that surfactant PVP can be dissolved in ethanol and improve the dispersibility of nanocarbon materials effectively [12,13,14] while make Mg isolate from air oxidizing. Our previous research showed that PCNS was surface-functionalized by dissolving PVP in ethanol, and the uniform PCNS with pristine structure could be obtained [15], suggesting that the PVP contributes to the preparation of nanocarbon-reinforced Mg composites with homogeneous nanocarbon dispersibility.

PVP is an organic reagent and plays a positive role in the dispersion of the reinforcing phase. However, it may have a negative effect on the mechanical properties of bulk Mg composites. To investigate the influence of PVP and the effectiveness of PCNS on Mg-based composite properties, PCNS was used as the reinforcing phase in this work. After ultrasonic pretreatment with dispersant PVP, PCNS was mixed with Mg powders. Then the composite powders were prepared by the PVP removal process. After this, these powders were fabricated into bulk Mg matrix composites by spark plasma sintered (SPS) and hot-press methods. The characterization and analysis of the samples were carried out to investigate the role of PVP in the composites preparation process and the influence of removing PVP on the properties of composites. Moreover, the possibility of PCNS for enhancing Mg was studied. The purpose of this work is to improve the preparation technology of homogeneously dispersed PCNS-reinforced Mg matrix composites and provide a reference for zero-dimensional carbon structure as reinforcement applied in the Mg matrix composites.

## 2. Materials and Methods

### 2.1. Materials

PCNS were selected as nanocarbon reinforcements, which were prepared by hydrothermal method at the laboratory [16]. The field emission scanning electron microscopy (FESEM) image of PCNS is shown in Figure 1. Mg powders (purity, 99%) with 75 μm of particle size and PVP with 30 K were supplied by Tianjin Guanfu fine chemical research institute Co., Ltd., Tianjin, China.

### 2.2. Material Processing

The preparation process of carbon-reinforced Mg composites used is illustrated in Figure 2.

#### 2.2.1. Preparation of Carbon-Reinforced Mg Composite Powders without PVP Pretreatment

A certain mass percentage of PCNS was mixed into 50 mL of ethanol. The ethanol slurry with the mixture was obtained by tip ultrasonic dispersion for 1 h. The mass percentage of the Mg powders corresponding to PCNS was put into ethanol slurry. Then, the PCNS-reinforced Mg composite powders, named PCNS/Mg, were obtained by removing ethanol with a rotary evaporator and vacuum drying.

#### 2.2.2. Preparation of Carbon-Reinforced Mg Composite Powders with PVP Pretreatment

PVP was dissolved in ethanol with a concentration of 10 g/L. A certain mass percentage of PCNS was added into the PVP solution of 50 mL, and their black slurry PCNS@PVP was obtained by tip sonicating for 1 h to functionalize the PCNS. Then, a mass percentage of the Mg powders corresponding to the reinforcing phase was put into PCNS@PVP slurry. Afterward, the rotary evaporator and vacuum drying were carried out to remove ethanol, for obtaining the primed Mg composite powders with PCNS uniformly dispersion, denoted as PCNS@PVP/Mg.

#### 2.2.3. PVP Removal Treatment

The PCNS@PVP/Mg primed composite powders were processed under flowing argon gas at 480 °C for 2 h to remove PVP, and thus the final PCNS@O/Mg composite powders were obtained.

#### 2.2.4. Processing of Carbon-Reinforced Mg Composite Bulks

Thirty-five grams of the composite powders was molded into a graphite mold of Φ40 mm × 15 mm by SPS at 50 kN and 560 °C. Then composite bulk was hot-pressed into a bar with Φ10 mm. The samples for the tensile tests were prepared by wire-electrode cutting.

The composite bulks, which are named as xPCNS/Mg for PCNS mass percentage, were fabricated with PCNS/Mg composite powders by SPS and hot-pressing. The samples with a different mass percentage of PCNS are shown in Table 1.

The composite bulks named as xPCNS@PVP/Mg were molded with PCNS@PVP/Mg composite powders by SPS and hot-pressing. In the meanwhile, the composite bulks tagged as xPCNS@O/Mg were fabricated by PVP removal treatment and molding. The samples with a different mass percentage of PCNS are shown in Table 2.

### 2.3. Characterization

The dispersion of reinforcement and fracture morphology of composite samples were examined by field emission scanning electron microscopy (FESEM, Japan Electronics Co. LTD, Tokyo, Japan). Phase analyses of composite powders were performed by X-ray diffraction (XRD, Rigaku Co. LTD, Tokyo, Japan), thermogravimetry-differential thermal gravity(TG-DTG, Beijing Optical instrument Factory, Beijing, China) and Fourier-transform infrared (FT-IR, Bio-Rad Laboratories, Hercules, CA, USA). The microstructures of samples were examined by optical micrographs. The interface structures of composites were investigated by energy-dispersive X-ray spectroscopy (EDS, Japan Electronics Co. LTD, Tokyo, Japan) and high-resolution transmission electron microscopy (HRTEM, Japan Electronics Co. LTD, Tokyo, Japan). The tensile test was carried out on a tension machine at room temperature.

## 3. Results and Discussion

### 3.1. Influence of PVP on Reinforcement Dispersion

Figure 3 shows the FESEM images of PCNS/Mg and PCNS@PVP/Mg composite powder samples. It can be observed in Figure 3a that the PCNS reinforcement were scattered on the cracks and surface of the Mg particles with obvious agglomeration. The functionalized PCNS by PVP pretreatment were homogeneously absorbed on the surfaces of the Mg particles without evident agglomeration of reinforcement in the cracks of composites, as shown in Figure 3b. The above results prove that the functionalization of carbon reinforcement by PVP could effectively improve the dispersibility of the reinforcing phases, which were absorbed on the surface of the Mg particles. This may have been due to the action of dispersant PVP, which increased the repulsion interactions of the electrical double layer, hydration retia and steric hindrance [17] and decreased Van der Waals attraction among reinforcement. When the ethanol evaporated, diffuse PCNS were coated on the surface of the Mg particles by PVP film.

### 3.2. Influence of PVP on Composite Powders

The XRD patterns of PCNS@PVP/Mg and PCNS@O/Mg composite powders were examined to analyze the transformation of composite powders in PVP removal. Figure 4a illustrates the characteristic peaks at around 32.16°, 34.39°, 36.59°, 47.80°, 57.36°, 63.05°, 68.65° and 69.99° corresponding to (100), (002), (101), (102), (110), (103), (112) and (210) of the Mg and a broad diffraction peak related to PCNS at around 22.1° corresponding to (002) [18]. As shown in Figure 4b, the low-intensity diffraction peak at around 43.0° was the characteristic peak of the MgO except for characteristic peaks of the Mg and PCNS in PCNS@O/Mg composites. This indicates that a small amount of the MgO was produced by oxidation of the Mg particles in PVP removal. It is worth noticing that the diffraction peak of PCNS in Figure 4b is sharper than Figure 4a. That can be concluded the PCNS of primed composite powders is graphitizing without destroying its structure in PVP removal [19]. The influence of PVP removal on properties of composite bulks should be tested by in latter.

In order to further study the transformation of composite powders after PVP removal. TG-DTG analysis was performed in an argon atmosphere in the temperature range from 30 to 500 °C at a heating rate of 5 °C/min. From the TG curves in Figure 5, it can be seen that the weight loss occurred from 290 to 470 °C, and the loss ratio was 2 wt.%. With the temperature of heating was below the melting point of the Mg, it was considered that the weightless is caused by the oxidization of PVP. As observed in the DTG curve, the weight loss peak at 330 °C was attributed to the hydroxides on the surface of the Mg particles on composite powders [20]. The loss weight peak at 420 °C was due to almost complete oxidation of PVP. It was proved that PVP removal could eliminate PVP and hydroxides on the surface of the Mg particles and produce a small amount of MgO at the same time.

### 3.3. Influence of PVP on Optical Micrographs of Bulk Composites

Figure 6 shows the optical micrographs of xPCNS@PVP/Mg composite bulks by SPS and hot-pressing. The left and right images are the microstructure of composites with vertical and parallel to the extruded direction, respectively. Moreover, the inset is the grain size distribution of the composites. The d_average_ represents the grain average size, which is the average width in the parallel section of the sample in order to reflect the microstructure of composite bulk objectively. As shown in Figure 6, the average grain sizes of the sample decreased with an increasing mass percentage of PCNS from 0.5 to 2 wt.%, the distribution range of grains decreased concurrently. The average grain size of 2PCNS@PVP/Mg was minimum with about 10.27 μm, and the size distribution range was the narrowest, from 3.21 to 43.57 μm, which showed the grains of 2PCNS@PVP/Mg were more small and uniform than those of other xPCNS@PVP/Mg samples. When the content of PCNS was added up to 4 wt.%, the average grain sizes and grain boundary width of samples are increased, with many clusters of PCNS observed at the Mg grains boundary obviously. The formation of broad grain boundary and voids may have been due to excessive PCNS can agglomerate so that they were not closely bound to the Mg matrix, resulting in corrosion and peeling in producing metallographic samples.

The samples of xPCNS@O/Mg showed the same trend as the xPCNS@PVP/Mg samples. As shown in Figure 7, the average grain sizes and size distribution range of xPCNS@O/Mg first decreased and then increased with increasing content of PCNS from 0.5 to 4 wt.%. The average grain size of 2 PCNS@O/Mg was minimum with about 12.41 μm, and the size distribution range was narrowest, from 3.34 to 30.20 μm, which proved that adding uniformly dispersed PCNS played a positive role in grain refinement and homogenization of composites. However, the grain sizes of xPCNS@O/Mg samples were larger than those of xPCNS@PVP/Mg in the same content. This was attributed to the increased grain sizes of the Mg particles after a high temperature of the PVP removal process.

In order to determine the positive impact of PVP in grain refinement, micrographs of xPCNS/Mg composites without PVP were observed. It can be seen in Figure 8 that the average grain sizes of xPCNS/Mg decreased with increasing mass percentage of PCNS from 0.5 to 1 wt.%, and many voids formed because of PCNS agglomeration. The average grain size of 1PCNS/Mg was 18.04 μm, larger than that of the composites prepared with PVP treatment. The volume of void increased with increasing content of PCNS; the width of the void section even reached 50 μm.

This illustrates the dispersity of PCNS is improved by PVP and still remains after the removal of the PVP. At the same time, the adding ratio of reinforcement can be increased by PVP treatment. All result proves PVP is effective in dispersing and binding PCNS in the Mg matrix. The dispersed PCNS can nail grain boundaries and hinder composite grain growth in the SPS process [21] and provides a valid nucleated particle for dynamic recrystallization in hot-pressing. The grains of composites are refined by hindering grain growth and dynamic recrystallization nucleation co-actions.

### 3.4. Interfacial Microstructure

The HRTEM and diffraction images at the interface of the 2PCNS@O/Mg sample are shown in Figure 9. There is a thick layer of about 6.2 nm at the interface between the PCNS and Mg matrix, as seen clearly in Figure 9a. The *d*-spacing value of this layer is measured as 0.214 nm corresponding to the (200) plane of the MgO. Combine with the EDS analysis of the sample (Figure 9c) and XRD pattern of PCNS@PVP/Mg and PCNS@O/Mg composite powders, confirms this layer is MgO film and produced in PVP removal. Furthermore, the planes with *d*-spacing values of 0.387 nm are exhibited in the interfacial region of 2PCNS@O/Mg, which are very close to 0.380 nm. The studies present that the interface-binding of two phases is strongest when the distance between graphene and MgO is 0.380 nm in research [22,23]. The PCNS has many physical and chemical properties similar to graphene for its graphite-like structure. As a result, the interface-binding is strong between MgO and PCNS in 2PCNS@O/Mg composites. In addition, the PCNS prepared by the hydrothermal method have good interface-binding due to the oxygen functional groups on the surface of PCNS are effectively promoting the interface-binding of two phases [24].

### 3.5. Tensile Behavior

There were three series of plate test specimens like “dog bone” with 30 mm length and 4 mm × 1.3 mm of initial cross-section area produced by xPCNS/Mg, xPCNS@PVP/Mg, xPCNS@O/Mg composites and pure Mg. The tensile testing of these specimens was applied in an AUTOGRAPH AGX-XD electronic universal testing machine under 20 kN of tensile load and 0.6 mm/min of the tensile rate at atmospheric temperature. Figure 10 shows the true tensile stress–strain curves of (a) xPCNS/Mg, (b) xPCNS@PVP/Mg, (c) xPCNS@O/Mg composites, and the value of the tensile yield, tensile ultimate strength and elongation of composite samples are shown in Table 3. The intensity of xPCNS@PVP/Mg and xPCNS@O/Mg composites are higher than that of xPCNS/Mg in the same mass percentage of reinforcement by analysis Figure 10 and Table 3, suggesting the PVP can disperse PCNS effectively, so that improve the properties of composites. The tensile yield and ultimate tensile strength of xPCNS@PVP/Mg and xPCNS@O/Mg composites first increase and then decrease with increasing content of PCNS from 0.5 to 4 wt.%. The best tensile property is presented at 2 wt.% of PCNS in the Mg matrix of these composites. When the content of PCNS is up to 4 wt.%, the tensile property of composite reduces because of the agglomeration of excessive PCNS, but larger than that of the Mg matrix. Both tensile yield (177 MPa) and ultimate tensile strength (206 MPa) of 2PCNS@O/Mg are the largest among these composites, increasing by 77% and 38% compared to that of pure Mg, respectively. The tensile yield strengths of these composites are improved accordingly, but their elongations decline compared with the Mg matrix. Combine with XRD pattern and HRTEM image analysis; this is suggested the MgO films are produced in the interface between PCNS and Mg matrix by the oxidization of PCNS oxygen functional groups and PVP with Mg after PVP removal process, as seen, the oxidation–reduction equation: Mg + O → MgO. It is worth to know, MgO is a typical oxide ceramic belonging to the brittle phase, which has good tensile strength and poor ductility. Jerng et al. [25] found the interfacial bonding between carbon and MgO was stable. Therefore, MgO films produced in two phases can improve tensile yield and ultimate tensile strength of composites but reduce the elongation simultaneously. Furthermore, the in situ synthesis of the MgO on the matrix surface enhances the dispersity of reinforcement in the matrix [26].

The comparison of xPCNS@PVP/Mg and xPCNS@O/Mg composites shows xPCNS@O/Mg composites obtained by removing PVP have better tensile properties than xPCNS@PVP/Mg composites. For example, the tensile yield, tensile ultimate strength and elongation of composites were improved after removed PVP at high temperature, which suggests the removal of PVP in composites has a positive effect on the properties.

The strength-to-weight ratio can be calculated according to the actual density and tensile yield strength of composites, are shown in Table 4. The strength-to-weight ratio of PCNS-reinforced Mg composites is obviously promoted with an increasing mass percentage of reinforcement because the density of PCNS is light, and the tensile strengths of composites are improved. The best strength-to-weight ratio presents at 2 wt.% of PCNS in the Mg matrix is 102.4 × 10^3^ N∙m/kg, with the increment of 78% larger than those of pure Mg. It still has a degree of advantage even compared with Al and carbon nanotubes (CNTs) or SiC-reinforced Mg composites [27] for the density of PCNS is lighter than that of CNTs and SiC.

## 4. Conclusions

The PCNS can effectively enhance the properties of the Mg-base, and the PCNS-reinforced Mg composites were successfully prepared by PVP pretreatment, the PVP removal process, spark plasma sintered and hot-pressed method. When ~2 wt.% of PCNS was added in the Mg matrix for preparing the composites, their tensile yield strength and specific strength exhibit 177 MPa and 102.4 × 103 N∙m/kg, which are 77% and 78% higher than those of the Mg matrix, respectively. The simple PVP pretreatment is effective for enhancing the properties of the Mg matrix composites due to the uniform dispersion of PCNS reinforcements in the Mg matrix. Then the tensile yield and ultimate tensile strength of composites can be further improved because MgO films between the matrix and reinforcement promote their interface compatibility after the PVP removal process. The findings suggest that the PCNS can effectively enhance the properties of the Mg matrix. However, the poor ductility of the MgO films and the zero-dimensional structure of PCNS cause a decrease in the elongation of composites.

## Figures and Tables

**Figure 1 nanomaterials-10-02281-f001:**
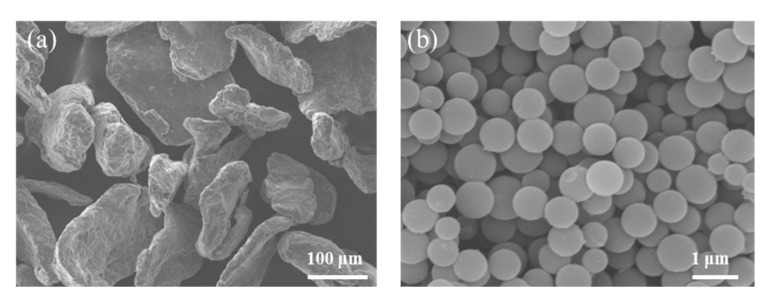
Field emission scanning electron microscopy (FESEM) image of (**a**) the Mg powders and (**b**) the microporous carbon nanospheres (PCNS).

**Figure 2 nanomaterials-10-02281-f002:**
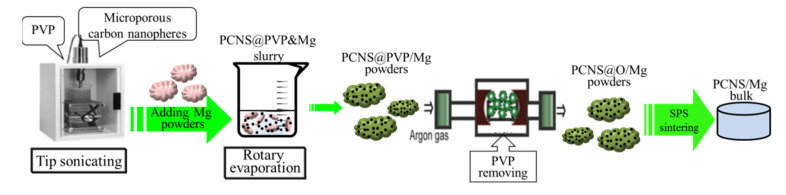
Preparation procedures for PCNS-reinforced Mg composite powders.

**Figure 3 nanomaterials-10-02281-f003:**
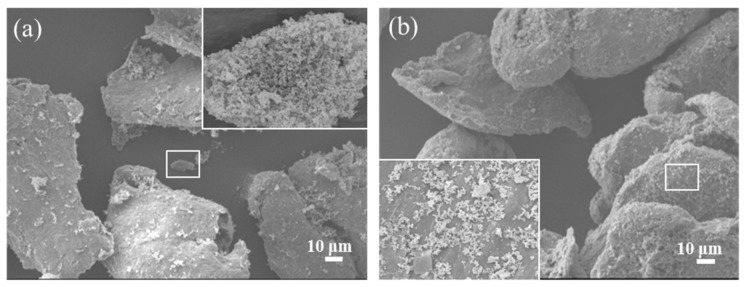
FESEM images of the samples by different decentralized processing: (**a**) PCNS/Mg, (**b**) PCNS@PVP/Mg.

**Figure 4 nanomaterials-10-02281-f004:**
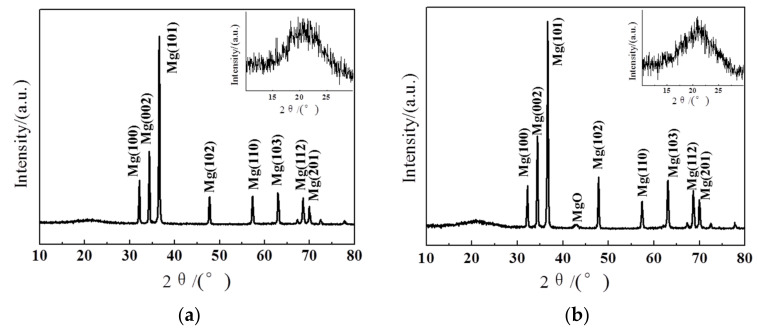
X-ray diffraction (XRD) pattern of PCNS-reinforced Mg composite powders before (**a**) and after (**b**) PVP removal.

**Figure 5 nanomaterials-10-02281-f005:**
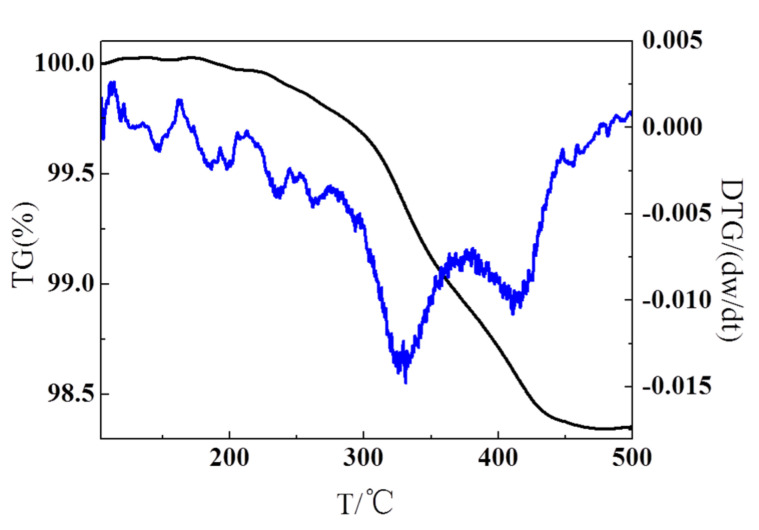
TG-DTG analysis of PCNS@PVP/Mg powder samples.

**Figure 6 nanomaterials-10-02281-f006:**
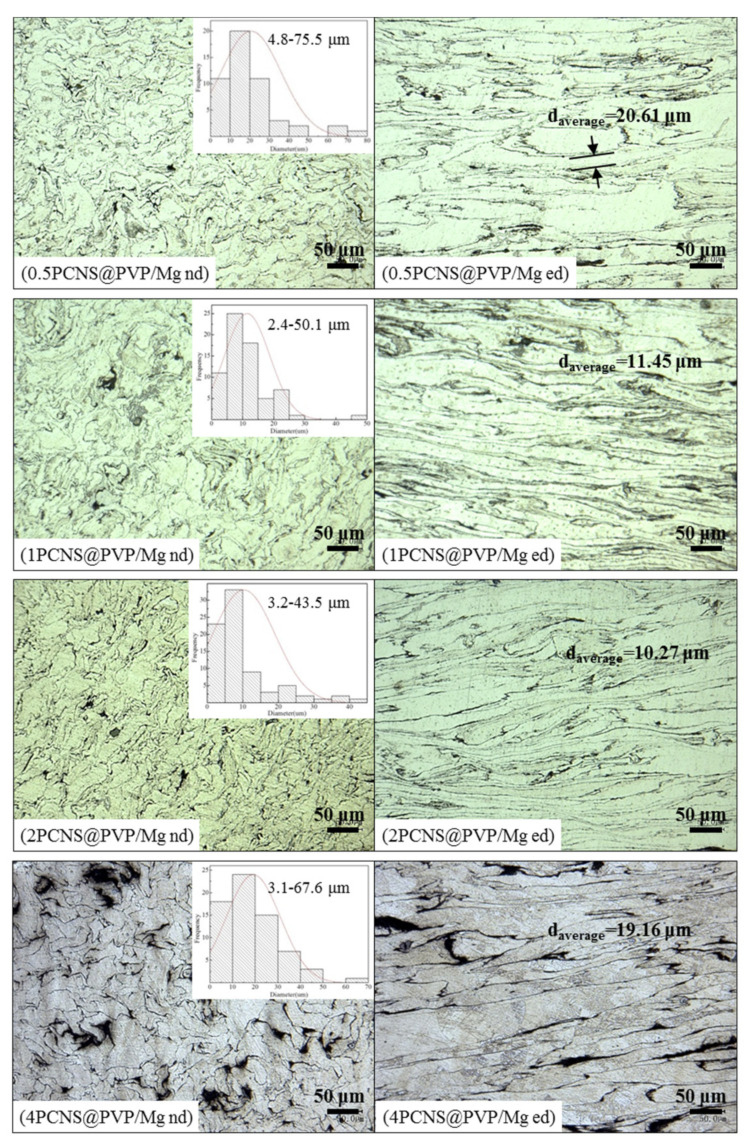
Optical micrographs of xPCNS@PVP/Mg bulk composites with the content of PCNS were 0.5, 1, 2 and 4 wt.%. The left and right images are the microstructure of composites with vertical (nd) and parallel (ed) to the extruded direction.

**Figure 7 nanomaterials-10-02281-f007:**
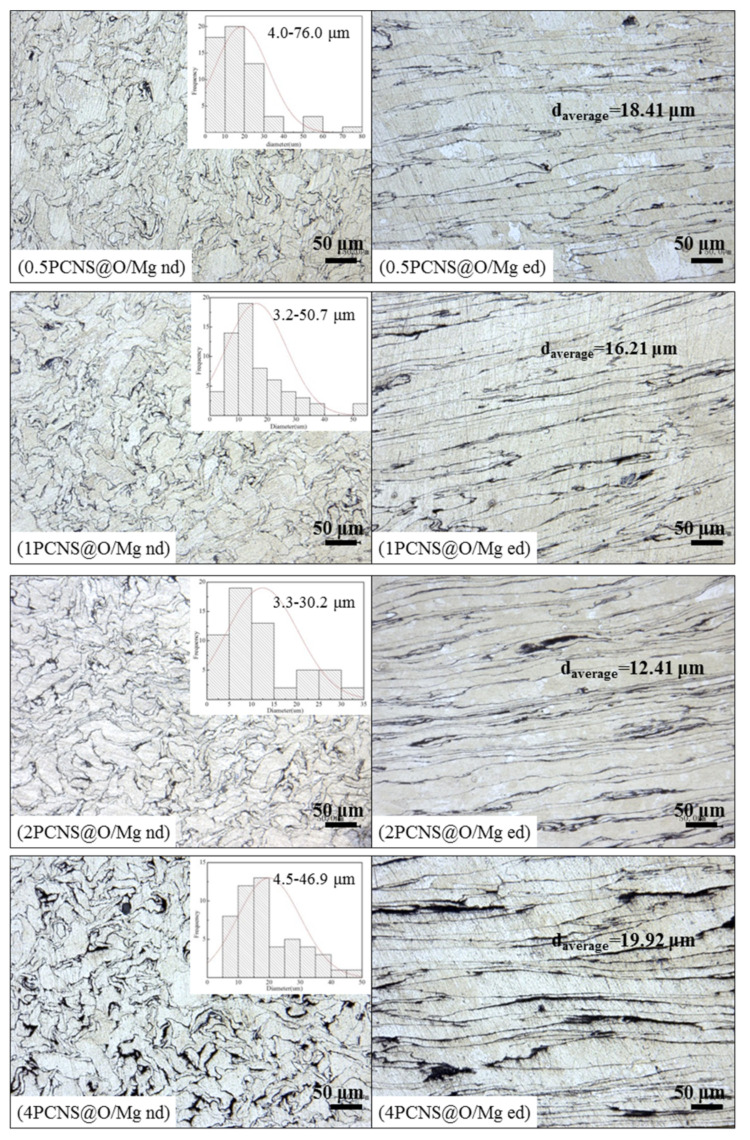
Optical micrographs of xPCNS@O/Mg bulk composites with the content of PCNS are 0.5, 1, 2 and 4 wt.%.

**Figure 8 nanomaterials-10-02281-f008:**
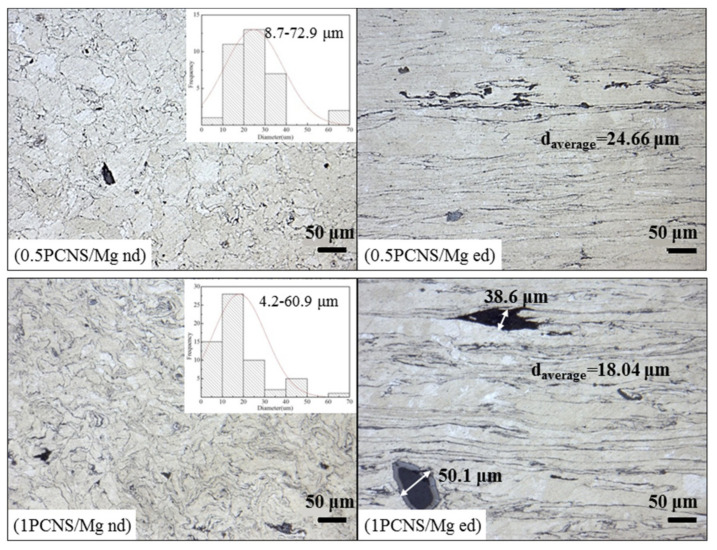
Optical micrographs of xPCNS/Mg bulk composites with the content of PCNS is 0.5 and 1 wt.%.

**Figure 9 nanomaterials-10-02281-f009:**
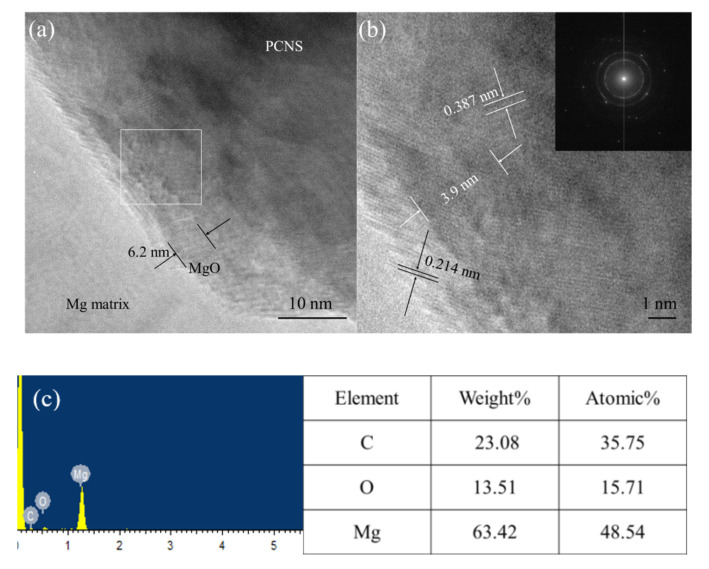
HRTEM image of (**a**) 2PCNS@O/Mg sample, (**b**) the enlarged image of the box in a and (**c**) the EDS results of the enlarged box b.

**Figure 10 nanomaterials-10-02281-f010:**
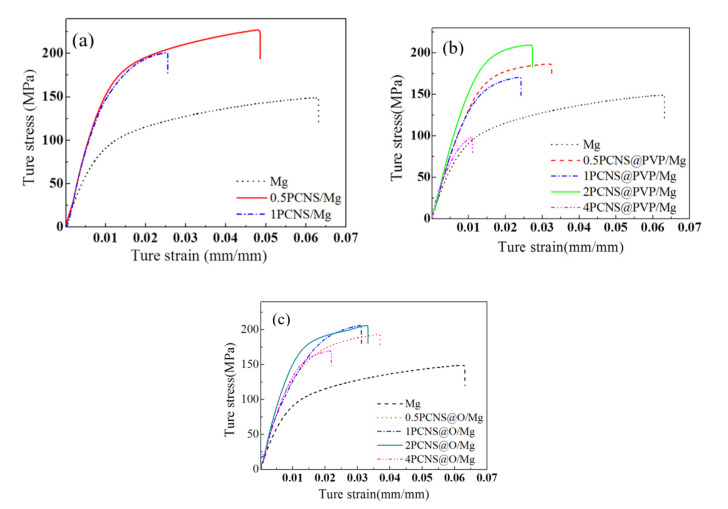
Tensile strength properties of (**a**) xPCNS/Mg, (**b**) xPCNS@PVP/Mg and (**c**) xPCNS@O/Mg composites.

**Table 1 nanomaterials-10-02281-t001:** Tags of samples without polyvinyl pyrrolidone (PVP) pretreatment by regular methods.

Mass Percentage of PCNS (wt.%)	Bulk Sample
0.5	0.5 PCNS/Mg
1	1 PCNS/Mg

**Table 2 nanomaterials-10-02281-t002:** Tags of PCNS/Mg composite samples with PVP pretreatment.

Mass Percentage of PCNS (wt.%)	Bulk Sample	Bulk Sample of Removing PVP
0.5	0.5 PCNS@PVP/Mg	0.5 PCNS@O/Mg
1	1 PCNS@PVP/Mg	1 PCNS@O/Mg
2	2 PCNS@PVP/Mg	2 PCNS@O/Mg
4	4 PCNS@PVP/Mg	4 PCNS@O/Mg

**Table 3 nanomaterials-10-02281-t003:** Tensile yield and ultimate tensile strength of PCNS-reinforced Mg composites.

Sample	Mass Percentage of PCNS (wt.%)	σ0.2 (MPa)	σUTS (MPa)	ε (%)
Mg	0	100	149	6.4
0.5 PCNS/Mg	0.5	140	227	4.8
1 PCNS/Mg	1	142	201	2.6
0.5 PCNS@PVP/Mg	0.5	150	186	3.2
1 PCNS@PVP/Mg	1	145	170	2.4
2 PCNS@PVP/Mg	2	165	209	2.7
4 PCNS@PVP/Mg	4	93	98	1.1
0.5 PCNS@O/Mg	0.5	148	193	3.7
1 PCNS@O/Mg	1	166	204	3.1
2 PCNS@O/Mg	2	177	206	3.4
4 PCNS@O/Mg	4	147	169	2.2

**Table 4 nanomaterials-10-02281-t004:** Strength-to weight ratio of PCNS-reinforced Mg composites.

Sample	Tensile Yield Strength (MPa)	True Density (g/cm^3^)	Strength to-Weight Ratio (N∙m/kg)
Mg	100	1.7362	57.6 × 103
0.5 PCNS@O/Mg	148	1.7381	85.2 × 103
1 PCNS@O/Mg	166	1.7319	95.8 × 103
2 PCNS@O/Mg	177	1.7282	102.4 × 103
4 PCNS@O/Mg	147	1.7101	86.0 × 103
Mg-1Al-0.3CNTs [27]	166	1.7443	95.2 × 103
Mg-1Al-0.6CNTs [27]	161	1.7400	92.5 × 103
Mg-1Al-0.6SiC [27]	156	1.7557	88.8 × 103
Mg-1Al-1.2SiC [27]	176	1.7646	99.7 × 103
Mg-1Al-2.4SiC [27]	186	1.7679	95.0 × 103

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
