# Peer review of "Effect of Surfactant Polyvinyl Pyrrolidone on the Properties of Microporous Carbon Nanospheres Reinforced Magnesium Matrix Composites"

_nanomaterials, 2020, doi:10.3390/nano10112281_

Round 1
Reviewer 1 Report
The composite preparation processes are described clearly. The microstructure is analyzed in detail. The results are new and very important.
Comments: 1) Scales in Figures 1, 3, 6, 7, 8, and 9 are missing.
2) The condition of tensile test (type of tensile test machine, shape and sample dimension, strain rate) should be given.
3) Could you give the individual contributions to the yield strengthening?
4) Reference 4 is not according to the Guide for Authors.
Author Response
Point 1: Scales in Figures 1, 3, 6, 7, 8, and 9 are missing.
Response 1: I have modified the scales of these diagrams according to the requirements.
Point 2: The condition of tensile test (type of tensile test machine, shape and sample dimension, strain rate) should be given.
Response 2: I gave the condition of tensile test in the first paragraph of “3.5 Tensile behavior”.
Point 3: Could you give the individual contributions to the yield strengthening?
Response 3: The individual contributions to the yield strengthening were given in the “3.5 Tensile behavior”.
Point 4: Reference 4 is not according to the Guide for Authors.
Response 4: For the reference 4 is an article published in Chinese magazine in 1996 without English title, I delete this reference.
Reviewer 2 Report
Review report of # nanomaterials-950191-peer-review-v1
Nanomaterials, Manuscript Number: #nanomaterials-950191-peer-review-v1, “Effect of surfactant polyvinyl pyrrolidone on the properties of microporous carbon nanospheres reinforced magnesium matrix composites”, by Lin Jin, Yong-Zhen Yang, Jian-Feng Fan, Bing-She Xu, used the PVP surfactant for the improvement of the dispersibility of carbon materials in magnesium (Mg) matrix. The manuscript should be accepted after few clarifications and corrections or should be added in the revised manuscript as mentioned below:
- The English corrections should be made thoroughly in the revised manuscript.
- The abstract must be focused on the base of research work not just a review way.
- The scale bar should be clear in SEM image.
- 2, the schematic illustration should be redesigned for clear understanding.
- All data and graphs should be replaced with new one and the analysis is not clear for readers.
Author Response
Point 1: The English corrections should be made thoroughly in the revised manuscript.
Response 1: I corrected the English mistakes in the revised manuscript.
Point 2: The abstract must be focused on the base of research work not just a review way.
Response 2: I made some amendments to the abstract of revised manuscript as required.
Point 3: The scale bar should be clear in SEM image.
Response 3: I have modified the scales of these diagrams according to the requirements.
Point 4: 2, the schematic illustration should be redesigned for clear understanding.
Response 4: The schematic illustration was redesigned and replaced the original Figure 2.
Point 5: All data and graphs should be replaced with new one and the analysis is not clear for readers.
Response 5: The Figure 4, 5 and 10 were instead of new graphs, and re-analyzed the data.
Round 2
Reviewer 1 Report
You present new and important results.